# The Deepfake Defense Stack: Why No Single Layer Works and How They Must Compose

## Abstract

Every defense against AI-synthesized media, whether passive detection, invisible watermarking, or content provenance, has been shown to fail when deployed in isolation. Detectors suffer 45–50% accuracy degradation from laboratory to deployment and collapse on outputs from unseen generator architectures. Watermarks are removable by regeneration attacks and screenshot capture. Provenance metadata is stripped by most social media platforms. Yet no prior work has formally analyzed how these defenses *compose*: which attack classes each layer blocks, where cascade failures propagate, and what residual vulnerabilities survive the full stack.

We present the first composition analysis of deepfake defenses. Through a Defense Composition Matrix, we map how detection, watermarking, and provenance hold up against seven attack classes, and we justify every cell against published evidence. The watermarking layer needs care. We separate marks stored in metadata from marks embedded in the content, because the two fall to opposite attacks. Once we separate them, only one attack still defeats every layer. That attack is the analog hole. Re-recording content from a screen breaks its link to the original capture, and no digital layer can follow it there. A second attack, infrastructure compromise, defeats the trust layer but leaves the other layers working, though degraded. Three further patterns emerge where stacking defenses creates a weakness that no single layer has on its own. We also state a Detection Ceiling Conjecture. It argues, with supporting evidence, that passive detection is approaching an information-theoretic limit. Provenance escapes that limit, but it pays a coverage cost of its own. Our analysis draws on the generation, detection, watermarking, and attack literature from 2014 to 2026. We do not survey any single layer in full; the surveys in Table 1 already do that. Our contribution is the cross-layer analysis they leave out, with enough background in Sections 3 to 5 to support it. We close with eight open problems, each stated as a falsifiable hypothesis with a proposed experiment.

## 1 Introduction

The AI-synthesized media ecosystem now generates content that is indistinguishable from authentic recordings under typical viewing conditions. A widely cited industry projection puts synthetic media circulating online at roughly 8 million files by 2025 DeepStrike (2025); deepfake-enabled fraud reached an estimated $1.65 billion in 2025, part of $2.19 billion in cumulative losses by early 2026 Surfshark (2026); and detection performance degrades by 45–50% from laboratory benchmarks to real-world deployment Chandra et al. (2025).

The response to this crisis has produced three classes of defense mechanisms. *Passive detection* builds classifiers that distinguish real from synthetic content based on pixel-level analysis. A second approach, *watermarking*, embeds imperceptible markers in AI-generated outputs to enable later identification. The third and most recent class, *content provenance*, takes a fundamentally different tack: rather than analyzing the content itself, it attaches cryptographic credentials at the point of capture to authenticate the origin of media.

Each class has been studied extensively in isolation. Detection methods spanning five paradigms have been surveyed by Pei et al. (2024), Croitoru et al. (2024), and at least 15 others since 2024. Watermarking has been systematized by Zhao et al. (2025) at IEEE S&P 2025. Proactive defenses (combining watermarking and disruption) are covered by Deng et al. (2025) in ACM Computing Surveys.

What no prior work has addressed is the question at the heart of deployment: *how do these defenses compose?* When detection fails against a new generator, does watermarking catch the content? When a watermark is stripped by a screenshot, does provenance survive? When an adversary targets all three layers simultaneously, which attack strategies succeed and which are blocked by the redundancy of the stack?

We provide the first composition analysis of deepfake defenses. Our contributions are:

1. A **Defense Composition Matrix** mapping seven attack classes against three defense layers, identifying which attacks each layer blocks, degrades, or is bypassed by (§7).

2. A **Cascade Failure Taxonomy** documenting how attacks that defeat one layer propagate through the stack, identifying the single attack class (the analog hole) that penetrates every layer and one (infrastructure compromise) that bypasses the primary trust layer (§7).

3. A **Detection Ceiling Conjecture** that ties a bound on how well any pixel-based detector can do to the way generator and real distributions converge, and sets it against the separate coverage limit that provenance faces (§8).

4. A **comparative taxonomy** of generation mechanisms across six modalities, detection across five paradigms, and proactive defenses, organized by the forensic signals each produces, as the technical basis for the composition analysis (§3–6).

5. A **gap analysis** positioning this work against 25+ competing surveys published since 2024 (§2).

6. Eight **open problems** with falsifiable hypotheses and proposed experimental protocols (§12).

Deng et al. (2025) provide the closest existing work: a multi-layer taxonomy covering detection, disruption, and authentication. We build on their taxonomy to analyze *composition*: how layers interact, where cascade failures occur, and which attacks survive when the layers are stacked together. Taxonomy organizes what exists; composition analysis tells you how to build a defense system.

## 2 Related Surveys and Positioning

At least 25 surveys published since 2024 cover portions of the terrain this paper addresses. Table 1 positions this work against the most relevant prior surveys across eight dimensions.

The key observation from Table 1 is that existing surveys cover at most four of the eight dimensions, and none addresses composition analysis or formal threat modeling across all three defense layers. The closest work, Deng et al. (2025), provides a multi-layer taxonomy covering detection, disruption, and authentication, but does not analyze how these layers interact under adversarial pressure or where cascade failures occur. Zhao et al. (2025) provide formal threat models for watermarking specifically but do not extend this analysis to the full defense stack. Our contribution fills the composition and formal analysis columns.

### 2.1 Search Strategy and Scope

We surveyed papers published between 2014 and March 2026 across arXiv, Semantic Scholar, Google Scholar, IEEE Xplore, and ACM Digital Library using queries combining "deepfake" with "detection," "provenance," "watermark," "C2PA," "adversarial," and "foundation model." For generation and detection methods (Sections 3–5), we prioritized papers at top-tier venues (NeurIPS, ICML, ICLR, CVPR, ICCV, ECCV, IEEE TPAMI, ACM CSUR). For proactive defenses (§6), we included industry standards (C2PA specifications), commercial products (SynthID, VideoSeal), and government guidance (NSA/CISA). For harm quantification (§11), we drew on government reports, verified incident databases (Surfshark, Sumsub), and investigative

Table 1: Positioning against related surveys. ✓ = covered in depth; ○ = mentioned but not primary focus; – = not addressed. No prior survey covers composition analysis or formal threat modeling across all defense layers.

| Survey | Year | Gen. | Det. | Prov. | WM | Gov. | Cross-mod. | Compos. | Formal |
|---|---|---|---|---|---|---|---|---|---|
| Tolosana et al. (Info. Fusion) | 2020 | ✓ | ✓ | – | – | – | – | – | – |
| Mirsky & Lee (2021) | 2021 | ✓ | ✓ | – | – | – | – | – | – |
| Pei et al. (ACM CSUR) | 2024 | ✓ | ✓ | – | – | – | – | – | – |
| Croitoru et al. (arXiv) | 2024 | ✓ | ✓ | – | – | – | ○ | – | – |
| Wang et al. (ACM CSUR) | 2024 | – | ✓ | – | – | – | – | – | – |
| Deng et al. (ACM CSUR) | 2025 | – | ✓ | ○ | ✓ | – | – | – | – |
| Zhao et al. (IEEE S&P) | 2025 | – | – | ○ | ✓ | – | – | – | ✓ |
| Zou et al. (arXiv) | 2025 | ○ | ✓ | – | – | ○ | – | – | – |
| Li et al. (ACM CSUR) | 2025 | – | ✓ | – | – | – | – | – | – |
| Gov. Info. Quarterly Survey | 2025 | ○ | ✓ | ✓ | ○ | ✓ | – | – | – |
| **This work** | **2026** | ✓ | ✓ | ✓ | ✓ | ○ | ✓ | ✓ | ✓ |

journalism, noting the provenance class of each statistic. We do not aim to cover any one field in full, and we do not claim to. We cite the landmark and representative works for each layer and attack class, enough to ground every cell of the matrix in published evidence.

## 3 Generation: A Comparative Taxonomy

We provide focused background on generation mechanisms, organized by the forensic signals each architecture produces and the defense layers each is vulnerable to. Comprehensive coverage of generation methods is available in Pei et al. (2024) and Croitoru et al. (2024); we summarize the properties relevant to our composition analysis.

### 3.1 Generative Adversarial Networks

The GAN framework (Goodfellow et al., 2014) introduced the minimax game between generator $G$ and discriminator $D$:

$$\min_G \max_D \ \mathbb{E}_{x \sim p_{\text{data}}}[\log D(x)] + \mathbb{E}_{z \sim p_z}[\log(1 - D(G(z)))] \tag{1}$$

A critical property follows directly from this formulation: the generator is explicitly optimized to produce outputs indistinguishable from real data. Any forensic classifier faces an adversary that has been specifically trained to defeat exactly such classifiers. Forensic adversariality is therefore an architectural feature of the GAN paradigm.

The architecture matured through three phases. Karras et al.'s Progressive GAN (Karras et al., 2018) introduced layer-by-layer resolution scaling from $4 \times 4$ to $1024 \times 1024$, enabling the first photorealistic face synthesis. STYLEGAN (Karras et al., 2019) introduced a mapping network $f : \mathcal{Z} \to \mathcal{W}$ that transforms a latent code into a disentangled intermediate space, with adaptive instance normalization injecting style at each synthesis layer. STYLEGAN2 (Karras et al., 2020) replaced AdaIN with weight demodulation, reducing the Fréchet Inception Distance (FID) to 2.84 on the Flickr-Faces-HQ (FFHQ) dataset at $1024^2$. STYLE-GAN3 (Karras et al., 2021) addressed "texture sticking" through formal application of the Nyquist-Shannon sampling theorem, achieving alias-free synthesis with continuous translation and rotation equivariance.

**Autoencoder-based face swapping.** The autoencoder paradigm, exemplified by DEEPFACELAB (Liu et al., 2023) (responsible for over 95% of deepfake videos with 35,000+ GitHub stars), uses a shared encoder and identity-specific decoders: $\hat{A}_B = D_A(E(f_B))$ where the target face $f_B$ is encoded through the shared encoder and decoded through the source decoder. SIMSWAP (Chen et al., 2020) introduced ID Injection for single-stage generalized swapping without per-identity training. HIFIFACE incorporated 3D Morphable Model supervision for shape-aware identity transfer at 1024 px.

**Face reenactment.** FACE2FACE (Thies et al., 2016) introduced real-time RGB-only reenactment via dense photometric tracking at 30+ fps. The First Order Motion Model (Siarohin et al., 2019) provided training-free animation through self-supervised keypoint learning with local affine transformations. The accessibility trajectory has been stark: what required PhD-level expertise in 2014 can now be accomplished with one-click mobile applications for approximately $10 per 50 videos.

**Detectability properties:** GANs leave systematic frequency-domain artifacts traceable to upsampling operations Frank et al. (2020), enabling spectral detection at 95%+ accuracy across multiple architectures. Face-swapping methods leave blending boundaries that Face X-Ray exploits. These artifacts are architecture-specific and do not transfer to diffusion models.

## 3.2 Diffusion Models and Text-to-Image Systems

Ho et al. (2020) established denoising diffusion probabilistic models. The forward process gradually adds Gaussian noise: $q(x_t \mid x_{t-1}) = \mathcal{N}(x_t; \sqrt{1 - \beta_t}\, x_{t-1}, \beta_t I)$, and a neural network $\epsilon_\theta(x_t, t)$ learns to reverse this process. The breakthrough for practical synthesis was Latent Diffusion (Rombach et al., 2022), which moved the process into a pre-trained autoencoder's latent space, reducing computation by approximately $64\times$ while adding cross-attention conditioning on text embeddings. Classifier-free guidance interpolates conditional and unconditional predictions with scale $s$: $\tilde{\epsilon}_\theta = \epsilon_\theta(\cdot, \varnothing) + s \cdot (\epsilon_\theta(\cdot, y) - \epsilon_\theta(\cdot, \varnothing))$.

This architecture underlies Stable Diffusion (open-source, August 2022, progressing from $512^2$ through SDXL to a Diffusion Transformer (DiT) backbone in SD 3), DALL-E 2 and 3 (Betker et al., 2023), and Midjourney (V1 through V7, adding video generation). The transition from GAN to diffusion generation has profound implications for detection. The diffusion process produces images through iterative refinement from noise, a mechanism that avoids the upsampling bottlenecks responsible for GAN spectral fingerprints.

**Detectability properties:** Diffusion models eliminate the spectral fingerprints that detectors relied on. Ricker et al. (2024) confirmed approximately 15.2 percentage-point drops in the area under the ROC curve (AUROC, or AUC) when GAN-trained detectors encounter diffusion outputs. DF40 Yan et al. (2024) found detectors at 85% AUC on face-swap content dropping to 47% on Stable Diffusion outputs. Detection now depends on semantic features rather than architectural artifacts, a shift that has driven the foundation model approaches discussed in §5.

## 3.3 Video Generation

Video synthesis reached a critical inflection in 2024–2025. Sora (Brooks et al., 2024) uses a DiT architecture operating on spacetime patches compressed by a spatial-temporal VAE, generating up to 20 seconds at 1080p (December 2024). Sora 2 (September 2025) added synchronized audio, persistent world state, and "Characters" (personalized avatars from short recordings) alongside a $1 billion Disney partnership. Google's Veo 2 (December 2024) and Veo 3 (May 2025) became the first models to natively generate synchronized dialogue, sound effects, and ambient audio alongside video. Runway released Gen-3 Alpha through Gen-4.5, with Gen-4.5 (December 2025) achieving top ranking on the Artificial Analysis Text-to-Video Leaderboard. Kuaishou's Kling AI iterated through 20+ versions, reaching Kling 2.6 with simultaneous audio-visual generation serving over 6 million users.

These capabilities now ship in consumer subscriptions: photorealistic video with native audio, persistent characters, and arbitrary text-directed control, produced from a text prompt.

**Detectability properties:** Temporal inconsistency, the signal exploited by recurrent detectors (FTCN, XcepTemporal), is progressively eliminated as video generators develop explicit temporal modeling. Frame-level detection via foundation models (GEND) averages per-frame softmax probabilities, discarding inter-frame information. Recent CVPR 2025 methods (DFD-FCG (Han et al., 2025), Spatiotemporal Adapter (Yan et al., 2025b)) begin to address temporal detection, but none has been evaluated on production-quality outputs from Sora 2 or Veo 3.

### 3.4 Voice Cloning and Lip Synchronization

Voice synthesis has progressed from autoregressive models (WaveNet, 2016; Tacotron 2, 2018, which reached a mean opinion score (MOS) of 4.53, against 4.58 for real speech) to zero-shot cloning. Microsoft's VALL-E (Wang et al., 2023) treats text-to-speech as conditional language modeling over discrete neural codec tokens, enabling voice cloning from 3 seconds of enrollment audio, trained on 60,000 hours of LibriLight. VALL-E 2 (Chen et al., 2024) achieved what researchers described as "human parity" on LibriSpeech and VCTK benchmarks, though the system was withheld from public release due to misuse risks.

ElevenLabs (Staniszewski & Dąbkowski, 2026) offers voice cloning from ∼30 seconds of enrollment audio in 70+ languages, and its technology has been directly implicated in deepfake fraud, including the Biden robocall of January 2024.

Wav2Lip (Prajwal et al., 2020) solved speaker-independent lip synchronization using a frozen pre-trained SyncNet discriminator, coupling arbitrary audio to any target face without per-subject training. Combined with voice cloning, this creates a complete audio-visual fabrication pipeline requiring only seconds of reference material.

**Detectability properties:** Synthetic speech leaves little at the signal level. Modern cloners reproduce the spectral envelope well, so detection falls back on prosody and meaning. Those cues fade as the models approach human quality. Lip synchronization is easier to catch. When audio drives the mouth, the timing between sound and lip shape slips, and detectors such as LipForensics read that slip. Even this signal narrows as generators improve, and recompression strips it early.

### 3.5 Neural 3D Synthesis

Mildenhall et al.'s NeRF (Mildenhall et al., 2020) represents scenes as continuous volumetric functions $F_\Theta : (\mathbf{x}, \mathbf{d}) \to (c, \sigma)$ rendered via differentiable volume integration. NeRF-based talking-head synthesis (AD-NeRF, ICCV 2021) extended this to audio-driven facial animation with full 3D consistency. Kerbl et al.'s 3D Gaussian Splatting (Kerbl et al., 2023) replaced implicit representations with explicit anisotropic 3D Gaussians and tile-based GPU rasterization, achieving real-time rendering at ≥100 fps at 1080p. This has been rapidly applied to talking-head synthesis: TalkingGaussian (ECCV 2024), GaussianTalker, and SyncGaussian (IJCAI 2025) all achieve real-time audio-driven avatars, enabling deepfake video calls that respond dynamically to conversation.

**Detectability properties:** Neural 3D methods render a face from a consistent geometric model, so they avoid many of the artifacts detectors look for in a single frame. The signals that remain are weak. There is occasional flicker between frames, and the shading can look wrong from some angles. Both improve as rendering quality rises. The output is a rendered image, not a face blended into a real photograph, so blending detectors such as Face X-Ray do not apply.

### 3.6 AI-Generated Text

Large language models represent a parallel but converging axis of AI-synthesized content. GPT, Claude, Gemini, and their successors generate text that is difficult for humans to distinguish from human-authored content. Hanley & Durumeric (2024) documented a 474% rise in machine-generated articles on misinformation sites between January 2022 and May 2023, against a 57.3% rise on mainstream news sites. Detection methods span watermarking (Kirchenbauer et al. (2023) embed token-level statistical signatures, though Pang et al. (2024) show these face fundamental trade-offs between robustness and spoofability), zero-shot statistical approaches (DetectGPT using log-probability curvature), and supervised neural classifiers. The convergence of text and media synthesis is critical: modern multimodal AI systems generate coordinated text, images, audio, and video from unified prompts, producing compound synthetic content that no single-modality detector can fully evaluate.

**Detectability properties:** Generated text carries no pixel signal at all. Detection rests on statistics of word choice, such as token watermarks and the curvature of model log-probabilities. These weaken as models

Table 2: Generation architectures and their detectability properties across six forensic dimensions. ✓ = signal present and exploitable; $\sim$ = signal weak or inconsistent; $-$ = signal absent.

| Architecture | Freq. artifacts | Temporal | Biological | Blending | Semantic | Metadata |
|---|---|---|---|---|---|---|
| GAN (face swap) | ✓ | $\sim$ | $\sim$ | ✓ | $\sim$ | ✓ |
| GAN (full synth.) | ✓ | $-$ | $-$ | $-$ | $\sim$ | ✓ |
| Diffusion (image) | $-$ | $-$ | $-$ | $-$ | $\sim$ | ✓ |
| Diffusion (video) | $-$ | $\sim$ | $\sim$ | $-$ | $\sim$ | ✓ |
| Voice cloning | $\sim$ | $-$ | $-$ | $-$ | ✓ | $\sim$ |
| Lip sync | $-$ | ✓ | ✓ | $\sim$ | ✓ | $\sim$ |
| NeRF/3DGS | $-$ | $\sim$ | $-$ | $-$ | $\sim$ | $\sim$ |
| LLM text | $-$ | $-$ | $-$ | $-$ | $\sim$ | $\sim$ |

match human writing, and a simple paraphrase can erase them. Of all the modalities, text is the hardest to catch after the fact, and the most dependent on the provenance and watermarking layers we analyze in §7.

### 3.7 Comparative Detectability Taxonomy

Table 2 maps each generation architecture to the forensic signals it produces, organized by the defense layer best positioned to detect it.

The taxonomy reveals a critical trend: as generation moves from GANs to diffusion and the newer modalities, the number of exploitable forensic signals falls. Modern diffusion-based systems leave only weak semantic signals and metadata traces. This pattern motivates our composition analysis (§7): if detection signals are diminishing, the defense stack must rely increasingly on provenance and watermarking layers that operate independently of pixel content.

## 4    Threat Model

Before assessing how each defense layer holds up, we define the adversary. We organize adversarial strategies against AI-synthesized media defenses into seven classes, ordered by increasing sophistication. These classes (A1–A7) are referenced throughout Sections 5–6 and form the rows of the composition analysis in §7.

**A1: Generator evasion.** Each new generation architecture produces outputs outside the training distribution of existing detectors. The GAN-to-diffusion transition eliminated the spectral fingerprints on which an entire class of detectors relied Ricker et al. (2024); Frank et al. (2020). This is not an adversarial attack in the traditional sense; it is the natural consequence of generator evolution.

**A2: Adversarial perturbation.** Imperceptible pixel-level modifications that flip detector predictions. Carlini & Farid (2020) reduced a forensic classifier from AUC 0.95 to 0.0005 with white-box attacks and to 0.22 in the black-box setting, perturbing as little as 1% of image area.

**A3: Compression and social media laundering.** Platforms apply proprietary recompression during upload, stripping high-frequency artifacts that detection methods exploit. XCEPTIONNET accuracy drops from 99.26% (raw) to 81.00% under heavy compression (c40).

**A4: Watermark removal and spoofing.** Zhao et al. (2024) proved that regeneration attacks remove 98% of invisible watermarks while keeping the peak signal-to-noise ratio (PSNR) above 30. LIGHTSHED Foerster et al. (2025) removes pixel-level protections from GLAZE and NIGHTSHADE with 99.98% success in 0.014 seconds per image. Spoofing is the other risk. Saberi et al. (2024) show that an attacker with only

black-box access to an image watermark can stamp a convincing mark onto any image. Real or harmful content then reads as watermarked.

**A5: Metadata stripping.** A screenshot removes all embedded metadata (C2PA credentials, EXIF data, invisible watermarks encoded in metadata fields) instantly. Platform re-encoding strips C2PA manifests unless the platform explicitly preserves them (as of March 2026, TikTok and Adobe tools do; most others do not).

**A6: The analog hole.** Re-recording content from a screen with a camera produces a new capture that carries no forensic connection to the original digital content. This is an irreducible attack that no digital defense can fully address, because it operates at the boundary between the digital and physical domains.

**A7: Infrastructure compromise.** Compromise of a C2PA certificate authority's signing key, a hardware Trusted Execution Environment (TEE) attestation root, or a watermark embedding key allows an adversary to produce fabricated content with valid credentials. This attack class does not exploit weaknesses in the defense *mechanism* but in the *infrastructure* on which the mechanism depends. We include it because our conclusion that provenance is the most resilient layer holds only under the assumption that provenance infrastructure is uncompromised; this assumption deserves explicit treatment.

## 5 Detection: Five Paradigms

We summarize detection across five paradigms, focusing on each paradigm's robustness under the attack classes defined in §4. Comprehensive surveys of detection methods are provided by Pei et al. (2024), Deng et al. (2025), and Zou et al. (2025); we extract the properties relevant to the composition analysis.

### 5.1 Temporal and Recurrent Architectures

Guëra & Delp (2018) established temporal consistency as a discriminating signal by feeding CNN frame features into an LSTM for video-level classification. Chintha et al. (2020) combined XCEPTIONNET with bidirectional LSTM/GRU layers and entropy-based cost functions for joint audio-visual detection, achieving 100% accuracy on FaceForensics++ (FF++) and strong cross-domain generalization. FTCN (Zheng et al., 2021) reduced all spatial kernels to $1 \times 1$ to force exclusive reliance on temporal features, achieving state-of-the-art cross-dataset generalization simultaneously across four benchmarks.

**Robustness under attack:** Temporal signals are vulnerable to A1 (generator evasion): video generators with explicit temporal modeling (Sora, Veo) are progressively eliminating inter-frame inconsistencies. They are robust to A3 (compression) at moderate levels but degrade under heavy recompression.

### 5.2 Spatial CNN and Attention Methods

Frame-level spatial analysis forms the backbone of the field. XCEPTIONNET (Chollet, 2017) (benchmarked by Rössler et al., ICCV 2019) achieves 99.26% accuracy on raw FF++ data through depthwise separable convolutions that efficiently capture fine-grained spatial artifacts. However, accuracy falls to 95.73% at moderate compression (c23) and 81.00% at heavy compression (c40), establishing that compression degrades detection systematically.

Face X-Ray (Li et al., 2020a) introduced a fundamentally different approach: detecting the universal blending boundary present in most face manipulations using an HRNet model trained *without any images from known manipulation methods*. This manipulation-agnostic design achieves 98.52% AUC on FF++ with 74.2% cross-dataset AUC on Celeb-DF. The key observation driving Face X-Ray is that most face manipulation methods share the common step of blending an altered face into an existing background image, and there exist intrinsic image discrepancies across blending boundaries regardless of the manipulation technique.

Multi-Attentional Detection (Zhao et al., 2021) recast detection as fine-grained classification using multiple spatial attention heads, a textural feature enhancement block, and attention-guided data augmentation, achieving ∼99% AUC on FF++ (c23).

Wang et al. (2020) demonstrated a remarkable property: a ResNet-50 trained only on ProGAN outputs generalizes at 92%+ AUC to 11 unseen CNN generators, suggesting shared low-level "CNN fingerprints" caused by upsampling artifacts. This property does not extend to diffusion models.

**Robustness under attack:** Spatial CNN detectors are highly vulnerable to A1 (28 pp cross- dataset drop from FF++ to Celeb-DF), A2 (Carlini & Farid (2020) reduce AUC to 0.0005 with white-box attacks), and A3 (18 pp drop under heavy compression).

### 5.3 Biological Signal and Frequency Methods

A third paradigm exploits physiological or physical signals that synthetic generators fail to reproduce. FAKE-CATCHER (Ciftci et al., 2020) uses remote photoplethysmography (rPPG), the subtle skin color changes from cardiac blood flow in the 0.7–4 Hz band, as authenticity descriptors, achieving 94.65% on FF++ and commercialized by Intel claiming 96% accuracy. However, Seibold et al. (2025) demonstrated that modern generators successfully reproduce rPPG signals, substantially undermining the assumption that biological signals are inherently unforgeable.

Li et al. (2018) exploited the observation that early deepfake training data rarely contained closed eyes, enabling a blink-frequency detector achieving 99% AUC on early datasets. This signal was rapidly nullified as generators incorporated natural blink patterns, illustrating the vulnerability of artifact-specific detection to generator adaptation.

Frank et al. (2020) (ICML 2020) demonstrated that all GAN architectures exhibit characteristic spectral artifacts in high-frequency DCT bands, traceable to nearest-neighbor or bilinear upsampling. A linear classifier trained on these spectral coefficients achieves over 95% accuracy across multiple GAN architectures. This represented the field's most productive forensic signal until the diffusion transition.

LIPFORENSICS (Haliassos et al., 2021) takes a semantic rather than artifact-based approach, pre-training a spatio-temporal network on lipreading across 500,000 utterances, then freezing the feature extractor and fine-tuning only the temporal classifier. Because the pre-training captures physiologically natural motion patterns, the method degrades more gracefully as generation quality improves.

**Robustness under attack:** Frequency signals are eliminated by the diffusion transition (A1; Ricker et al. (2024) confirmed 15.2 pp AUROC drop). Biological signals (rPPG, blink patterns) are being systematically reproduced by modern generators. Semantic approaches (LIPFORENSICS) are more resilient but still degrade on high-quality generators. All three signal types are vulnerable to A3 (compression strips high-frequency information that all three exploit).

### 5.4 Vision Transformer and Hybrid Architectures

Coccomini et al. (2022) combined EFFICIENTNET-B0 with a Vision Transformer, achieving AUC 0.951 on DFDC. ViT-based temporal attention over frame embeddings outperforms both pure-CNN and pure-RNN approaches on challenging data.

**Robustness under attack:** These hybrid detectors keep a convolutional stem, and they inherit its weaknesses. They degrade under a new generator (A1) and under compression (A3), much as a pure CNN does. Their robustness to adversarial perturbation (A2) has not been tested in any systematic way. The transformer attention adds capacity, but it adds no new defense against the attacks in §4.

### 5.5 Foundation Model Approaches

The most promising recent direction adapts large-scale pre-trained vision or vision-language models for detection. The turning point came when Ojha et al. (2023) showed that a linear classifier on top of unmodified CLIP representations could generalize to unseen generators, a result that suggested pre-trained vision encoders already capture the distinction between real and synthetic content at a coarse level.

This result catalyzed a wave of adaptation strategies. CLIPping the Deception (Khan & Dang-Nguyen, 2024) adapts CLIP via parameter-efficient prompt tuning, outperforming prior methods by +5.01% mean average

Table 3: Detection performance summary (AUC %). FF++ = FaceForensics++ HQ; Cross = FF++ → Celeb-DF v2. Values are as reported by each method's authors under that paper's own training and evaluation protocol, not a head-to-head re-evaluation; the Cross column reflects the closest comparable setting. The numbers are indicative of the trend, not a controlled comparison.

| Paradigm | Method | FF++ | Cross | Venue |
|----------|--------|------|-------|-------|
| Spatial CNN | XceptionNet | 95.7 | 71.5 | ICCV'19 |
| Spatial CNN | Face X-Ray | 98.8 | 74.2 | CVPR'20 |
| Temporal | FTCN | 98.8 | 86.9 | ICCV'21 |
| Biological | LipForensics | 97.1 | 82.4 | CVPR'21 |
| ViT Hybrid | ENet+ViT | 95.1 | 64.8 | ICIAP'22 |
| Foundation | CLIPping | 96.7 | 88.1 | ICMR'24 |
| Foundation | GenD (CLIP) | 96.0 | 92.8 | 2025 |

precision (mAP) across 21 datasets while using less than one-third of training data. GEND (Yermakov et al., 2025) tunes only the Layer Normalization parameters of CLIP ViT-L/14 (0.03% of weights), enforces L2 normalization, and applies uniformity and alignment losses, achieving state-of-the-art average cross-dataset AUROC across 14 benchmarks spanning six years of deepfake evolution. $D^3$ (Yang et al., 2025) scales multi-generator training with a parallel discrepancy branch. EFFORT (Yan et al., 2025a) decomposes weight matrices via singular value decomposition (SVD), freezing principal components and fine-tuning only the residual orthogonal subspace.

At the multimodal LLM frontier, several 2025–2026 methods integrate large language models with detection. FakeVLM (Wen et al., 2025) trains on 100,000+ images with natural-language artifact annotations, producing both a classification and a human-readable explanation of the observed artifacts. VIGIL (Li et al., 2026) takes a forensic-inspired approach: it decomposes the face into anatomical regions, independently examines each region for evidence of manipulation, and synthesizes a verdict from the accumulated part-level findings. ForensicZip (Lai et al., 2026) tackles the computational bottleneck of processing high-resolution content through multimodal LLMs, achieving a 2.97× speedup by selectively retaining tokens that carry forensic rather than semantic information.

A companion paper examines why foundation models generalize, testing three competing hypotheses, and where they fail.

**Robustness under attack:** Foundation models show improved robustness to A1 (new generators) compared to CNN detectors, likely because they exploit semantic rather than artifact-specific features. Their robustness to A2 (adversarial perturbation) has not been systematically evaluated; CLIP is known to be vulnerable to typographic and embedding-space attacks, and VLM-based detectors face additional prompt injection risks. We flag adversarial evaluation of foundation-model detectors as the most urgent open experimental question.

## 5.6 Cross-Paradigm Summary

Table 3 summarizes representative performance across paradigms.

The reported numbers trend upward across paradigms, from spatial CNN methods (71.5%) through temporal (86.9%) and semantic (82.4%) approaches to foundation models (92.8%). Because these figures come from different papers and protocols, we read the trend as indicative rather than as a controlled comparison; even so, its direction is consistent, with foundation models narrowing the cross-dataset gap from roughly 28 points (XCEPTIONNET) to a few points (GEND). The challenge of entirely new generation paradigms remains open.

# 6 Proactive Defenses: Provenance and Watermarking

We survey three classes of proactive defense, each analyzed for its vulnerability profile under the attack classes defined in §4.

## 6.1 C2PA Content Provenance

The Coalition for Content Provenance and Authenticity (C2PA, founded February 2021 by Adobe, Arm, BBC, Intel, Microsoft, and Truepic) has published specification v2.2 (May 2025). Content Credentials are cryptographically signed manifests that bind public-key signatures, content hashes, and trusted timestamps to the media itself. Any tampering with the content breaks the cryptographic signature. The Content Authenticity Initiative (CAI), C2PA's implementation arm, surpassed 5,000 members by mid-2025, including TikTok, Meta, Google, OpenAI, Cloudflare ($\sim$20% of web traffic), and major news organizations (AP, AFP, Reuters, Washington Post).

C2PA's critical advantage is architecture-agnostic authentication: it makes no assumptions about how content was generated and is not subject to obsolescence as generation methods evolve. This is the property that makes it robust to attack classes A1–A4 in the composition matrix.

TikTok implemented C2PA Content Credentials at scale (May 2024) and removed 51,618 synthetic media videos in H2 2025. Meta labels AI content using C2PA metadata detection, but its Oversight Board ruled in March 2026 that the system "falls short," finding only $\sim$30% of AI content correctly labeled.

**Vulnerability profile:** C2PA is robust to attacks A1–A4 (its cryptographic integrity is independent of pixel content, watermark status, and generator type). It is vulnerable to A5 (metadata stripping: platforms that re-encode content strip the C2PA manifest), A6 (analog hole), and A7 (infrastructure compromise: CA key theft or fraudulent device attestation). The critical adoption gap: most intermediary platforms still strip C2PA metadata during upload.

## 6.2 Hardware-Anchored Provenance

Qualcomm's Snapdragon 8 Elite Gen 5 (September 2025) (Truepic, 2025) embeds Truepic's Secure Media Library in the mobile TEE, bringing hardware-certified provenance to billions of devices via Samsung Galaxy S26 and Xiaomi 17. Hardware-origin credentials are structurally superior to software-added ones: they are embedded at capture time within a tamper-resistant boundary before any software manipulation is possible.

## 6.3 AI Output Watermarking

SYNTHID (Google DeepMind, 2023) has watermarked over 10 billion items across images (Imagen), video (Veo), audio (Lyria), and text (Gemini). OpenAI adds C2PA metadata to all DALL-E 3 outputs with $\sim$98% detection accuracy at <0.5% false positive rate. VIDEOSEAL (Fernandez et al., 2024) (December 2024) provides open-source video watermarking, later expanded to the comprehensive Meta Seal framework covering all modalities. Kirchenbauer et al. (2023) introduced token-level statistical watermarking for LLM text at ICML 2023, modifying logit distributions during generation to embed detectable signatures.

**Vulnerability profile:** Watermarks are robust to A1 (new generators do not affect embedded marks) and partially robust to A3 (moderate compression). They are vulnerable to A4: Zhao et al. (2024) (NeurIPS 2024) proved that regeneration attacks remove 98% of invisible watermarks while maintaining PSNR >30. LIGHT-SHED Foerster et al. (2025) (USENIX Security 2025) demonstrated generalizable removal of pixel-level protections from GLAZE, NIGHTSHADE, and similar tools with 99.98% success in $\sim$0.014 seconds per image. Spoofing is the other risk. Saberi et al. (2024) show that an attacker with only black-box access can stamp a convincing mark onto any image, so real or harmful content reads as watermarked. The harder a mark is to remove, the easier it tends to be to forge. The simplest attack (a screenshot) removes all metadata-based watermarks instantly. These results indicate that watermarking is a necessary but insufficient layer requiring complementary mechanisms.

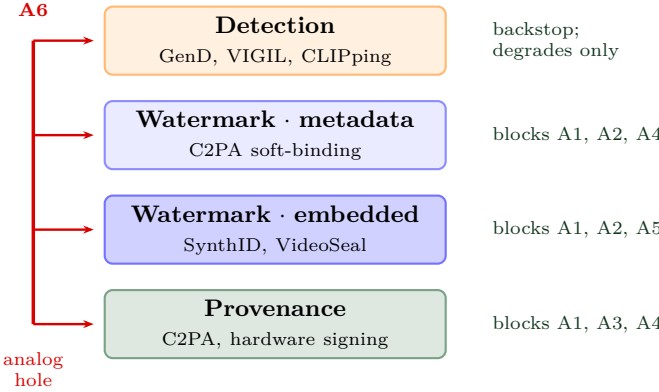

*A7: forged credentials bypass provenance (the trust layer)*

Figure 1: The composed defense stack, matching Table 4. Each layer blocks a different subset of attacks (right), so together they cover A1 through A5: every one is blocked by at least one layer. Splitting watermarking into metadata and embedded marks is what closes the gap, because regeneration (A4) and metadata stripping (A5) defeat opposite mark types. Only the analog hole (A6) pierces every layer at once, the single irreducible bypass. Infrastructure compromise (A7) forges valid credentials and so bypasses provenance, the trust layer, though the content layers still degrade it.

## 6.4 Consent Infrastructure

HaveIBeenTrained (Spawning, 2023) processed over 1 billion opt-outs. Nightshade and Glaze poison training data and mask artistic style, but LightShed's circumvention of both tools indicates that perturbation-based protection faces the same arms-race dynamics as detection.

# 7 The Defense Composition Matrix

The individual defense mechanisms surveyed in Sections 5 and 6 are well understood in isolation. What has not been analyzed is how they interact when deployed as layers of a unified defense stack. We present this composition analysis here, summarized in Figure 1.

## 7.1 The Composition Matrix

Table 4 maps the seven attack classes of §4 against the defense layers. We split the watermarking layer in two, because metadata marks and embedded marks fail to opposite attacks and cannot share one rating. We use four ratings. **Blocked (B)**: the layer keeps more than 90% of its laboratory performance against the attack. **Degraded (D)**: performance drops by 15 to 50%, but the layer keeps some value. **Bypassed (X)**: performance falls to near chance (below 60% AUC), or the attack disables the layer. **Not applicable (N/A)**: the attack does not act on the signal the layer reads. Every non-trivial rating is tied to a published datapoint in Table 5. The ratings rest on image and video evidence, where the literature is deepest. We treat audio, 3D, and text qualitatively in §3 and do not rate them here. Their evidence is still too thin to defend a judgment for each cell.

## 7.2 Cascade Failure Analysis

The matrix reveals several structural properties of the defense stack.

**Provenance is the most resilient layer.** It survives attacks A1 through A4, because a cryptographic signature does not depend on the pixels or on the watermark. Its weakness is not manipulation but coverage. Metadata stripping (A5), the analog hole (A6), and infrastructure compromise (A7) all defeat it, and a missing credential tells us nothing (§8).

Table 4: Defense Composition Matrix. B = blocked (>90% of laboratory performance retained); D = degraded (15–50% drop, partial value); X = bypassed (<60% AUC or structurally inoperative); N/A = attack does not act on this layer's signal. We separate metadata marks ($WM_{meta}$) from marks embedded in the content ($WM_{emb}$, for example SYNTHID and VIDEOSEAL); the two fail to opposite attacks. Only the analog hole (A6) bypasses every layer, shown in  red . Table 5 justifies each rating.

| Attack Class | Detect. | $WM_{meta}$ | $WM_{emb}$ | Prov. | Composition |
|---|---|---|---|---|---|
| A1: Generator evasion | D | B | B | B | 3 block, 1 degrade |
| A2: Adversarial perturb. | X | B | B | N/A | 2 block; det. bypassed |
| A3: Compression | D | D | D | B | 1 block, 3 degrade |
| A4: Watermark removal | D[†] | B | X | B | 2 block, 1 bypass |
| A5: Metadata strip | D | X | B | X | 1 block, 1 degrade, 2 bypass |
| A6: Analog hole | X | X | X | X | **All layers bypassed** |
| A7: Infra. compromise | B→D[‡] | D | D | X | **Primary trust layer bypassed** |

[†] Regeneration that removes a watermark may leave residual artifacts that detection can still exploit. [‡] Pixels remain synthetic, so detection is nominally unaffected (B); but deployments that trust credentials short-circuit detection on credentialed content, degrading its effective coverage (D).

Table 5: Per-cell justification for the non-trivial ratings in Table 4. Trivial cells (a layer blocking an attack its signal is immune to) are omitted.

| Cell | Rate | Basis |
|---|---|---|
| Detect. / A1 | D | CNN detectors collapse (28 pp FF++→Celeb-DF; 85→47% on diffusion Yan et al. (2024)), but foundation-model detectors degrade and survive (92.8% cross-dataset Yermakov et al. (2025)); truly novel architectures remain untested. |
| Detect. / A2 | X | AUC 0.95→0.0005 (white-box), 0.22 (black-box) Carlini & Farid (2020). |
| Detect. / A3 | D | XCEPTIONNET 99.26→81.00% at c40 Rössler et al. (2019). |
| Detect. / A7 | B→D | Synthetic pixels remain detectable, but credential-trusting pipelines bypass detection on credentialed content. |
| $WM_{meta}$ / A4 | B | No signal-domain payload to regenerate away; metadata marks survive pixel regeneration. |
| $WM_{meta}$ / A5 | X | Screenshot / platform re-encode strips metadata fields. |
| $WM_{emb}$ / A4 | X | Regeneration removes 98% of pixel watermarks at PSNR >30 Zhao et al. (2024). |
| $WM_{emb}$ / A5 | B | SYNTHID/VIDEOSEAL are signal-domain and survive screenshot and re-encoding by design Google DeepMind (2023); Fernandez et al. (2024). |
| WM / A3 | D | Heavy recompression degrades both mark types. |
| WM / A7 | D | Embedding-key compromise enables spoofing; image watermarks can be forged with only black-box access Saberi et al. (2024). |
| Prov. / A1–A4 | B | Cryptographic signature is independent of pixels, marks, and generator. |
| Prov. / A2 | N/A | Pixel perturbation cannot alter a cryptographic signature. |
| Prov. / A5 | X | Platforms strip C2PA manifests on re-encode Nemecek et al. (2026). |
| Prov. / A6 | X | Re-captured content carries no credential. |
| Prov. / A7 | X | A forged signing key yields valid credentials on fabricated content. |

**The two watermark classes cover each other.** A single watermark column hides this. Metadata marks and embedded marks are defeated by opposite attacks. Regeneration (A4) removes an embedded mark but leaves a metadata mark in place. Metadata stripping (A5) does the reverse, removing the metadata mark

but leaving the embedded mark, which SynthID and VideoSeal are built to survive. Deploy both, and each covers the other's gap.

**Detection and watermarking fail to different attacks.** The attacks that beat detection, a new generator (A1) or an adversarial perturbation (A2), do not touch a watermark, because the mark is embedded before the attack reaches the content. The attacks that beat a watermark, regeneration (A4) or a metadata strip (A5), do not beat detection, because the removal can leave its own trace. Detection and both watermark types together are far harder to defeat than any one alone.

**Only the analog hole defeats the whole stack.** Once the watermark layer is split, one attack still beats every layer. It is the analog hole (A6). Re-recording from a screen breaks the link to the original capture, so detection, both watermarks, and provenance all fail together. No digital defense can close it. Two attacks come close but do not. Metadata stripping (A5) looks like a full bypass until you separate the watermark types, since the embedded mark survives it. Infrastructure compromise (A7) defeats the trust layer but leaves the other layers standing, if weakened.

**Compression wears down every content layer.** Compression (A3) defeats no layer outright, but it degrades detection and both watermark types at once. A platform that recompresses hard on upload weakens all three content defenses together. Provenance is the exception, since it does not read the pixels.

### 7.3  Answering the Motivating Questions

The matrix lets us return to the three questions that opened this paper and answer each directly.

**When detection fails against a new generator, does watermarking catch the content?** Yes, if the content was marked when it was made. A new architecture (A1) defeats CNN detectors and weakens the rest, because detection keys on traces left by a particular generator. It does not touch a mark already embedded in the output. Both watermark types still hold. The catch is conditional. Watermarking helps only where the generator embedded a mark to begin with.

**When a watermark is stripped by a screenshot, does provenance survive?** No. The intuitive answer is wrong here. A screenshot is a metadata strip (A5). It removes the C2PA manifest and any metadata mark, so provenance does *not* survive. The embedded watermark does survive, because it lives in the pixels, not the metadata. So the fallback against a screenshot is the embedded mark, not provenance. That reverses what a layered view of trust would suggest.

**When an adversary targets all three layers at once, what succeeds and what is blocked?** Only the analog hole (A6) succeeds against the full stack. Infrastructure compromise (A7) defeats the trust layer, but it leaves the other layers degraded rather than gone, and every other attack is blocked by at least one layer. The redundancy is real, but it is not even. It is strongest against the content attacks (A1 through A4) and weakest at the line between the digital and the physical, where no digital layer reaches.

### 7.4  Emergent Composition Effects

We identify three cases where stacking defenses creates effects that are absent from individual layers.

**E1: Authenticated contradictions.** When content carries valid C2PA provenance *and* a watermark indicating AI generation, the two signals contradict each other: provenance says "captured by a real camera" while the watermark says "generated by AI." This occurs when a real photograph is post-processed by an AI system that embeds a generation watermark. The "Authenticated Contradictions" paper Nemecek et al. (2026) shows that this conflict is not hypothetical; it arises naturally in editing workflows and creates a trust ambiguity that individual layers do not.

**E2: False provenance amplification (hypothesis).** If an adversary compromises a C2PA signing key (e.g., through a supply-chain attack on a camera manufacturer), they can produce fabricated content with valid provenance. We hypothesize that in a stacked system where users have learned to trust provenance as the primary signal, the damage from compromised provenance is amplified relative to a system without provenance, because the presence of credentials may train users to lower their guard. This hypothesis

requires empirical validation through user studies comparing trust calibration in provenance-present versus provenance-absent environments; we include it as an open question rather than a demonstrated finding.

**E3: Detection-watermark feedback loops (hypothesis).** If detection methods use the presence or absence of watermarks as an input feature, an adversary who learns to embed legitimate watermarks into fabricated content (a spoofing attack in the sense of Saberi et al. (2024)) could exploit this dependency to fool both the watermark verifier and the watermark-aware detector simultaneously. We are not aware of deployed detection systems that explicitly condition on watermark status, but the architectural possibility exists as watermark-aware detection pipelines are proposed. We flag this as a design risk for future systems rather than a demonstrated vulnerability.

## 8 The Detection Ceiling

We argue that passive detection faces a structural limit that provenance does not. Provenance has a different limit, which we come to below. The detection limit has two parts. One is a textbook inequality, stated next. The other is an empirical conjecture about how generators evolve. We keep them apart, because the inequality is proved and the conjecture is not.

**Observation (balanced-accuracy ceiling).** For any binary classifier $D$ distinguishing samples from distributions $p_G$ and $p_{\text{real}}$ under equal priors, the *balanced accuracy* satisfies

$$\text{Acc}_{\text{bal}}(D, G) \leq \frac{1}{2} + \frac{1}{2} \text{TV}(p_G, p_{\text{real}}). \tag{2}$$

This is a standard result, not ours. It follows from the Neyman-Pearson lemma and Scheffé's identity, equivalently Le Cam's two-point method, under equal priors. We state it for balanced accuracy on purpose. The bound governs a single decision threshold. It does *not* carry over to AUC. AUC measures ranking across all thresholds, and it can exceed the right-hand side even when the bound on balanced accuracy is tight. So we keep the formal claim in balanced accuracy. The AUC numbers below illustrate the same trend, but the inequality does not bound them.

**Conjecture 1** (Detection Ceiling). *As generators improve, $TV(p_G, p_{real}) \to 0$, and the balanced accuracy of any detector that reads only the pixels falls toward chance ($Acc_{bal} \to 0.5$). Provenance is not subject to this ceiling, because it reads a cryptographic credential, not the pixels. It faces a different limit on coverage, which we discuss below.*

The inequality is settled. The conjecture is not. The conjecture claims that TV is shrinking as generators improve. We can only test it indirectly, because TV between high-dimensional image distributions cannot be estimated from a finite sample.

Agarwal & Varshney (2019) made this argument for deepfake forensics, bounding detection error from a robust hypothesis-testing view. Our contribution is not the inequality, which is textbook. It is where the inequality sits. Detection is bounded by a quantity that shrinks as generators improve. Provenance is not. That is why the two belong together, as complementary layers rather than substitutes. The record fits a shrinking margin. Reported cross-dataset AUC for GAN-trained detectors falls from over 95% on GAN content to about 47% on diffusion content. We read this as a sign of the trend, not as a measurement of TV.

We must separate an information limit from a limitation of today's detectors. The 47% may only mean that GAN-trained detectors overfit to GAN traces, not that $p_G$ and $p_{\text{real}}$ have truly converged. Foundation-model detectors reach 92.8% cross-dataset, which suggests the current ceiling is a matter of architecture, not information. If the conjecture holds, better generators (Sora 2, Veo 3) would lower the bound even for these detectors. The open question would then be when the ceiling binds, not whether. For now we cannot tell this path apart from a lasting architectural plateau, and we do not claim to.

**Supporting evidence.** The empirical record is consistent with this conjecture. Two further results fit the conjecture. Deepfake-Eval-2024 Chandra et al. (2025) reports a 45 to 50% drop in detection performance from the laboratory to in-the-wild deployment, measured across 44 hours of real-world content. Wang et al. (2025), a recent preprint, fits detection error to a power law $1 - \text{AUC} = A \cdot N^{-\alpha}$ in training diversity. It gives no guarantee that the law holds for a new kind of generator.

**Why provenance escapes this ceiling, and the ceiling it faces instead.** Provenance reads a different signal. A C2PA credential is a cryptographic claim about where the content came from, for example that device X captured it at time T. It does not depend on the pixels, so the convergence of $p_G$ toward $p_{\text{real}}$ leaves it untouched. On its own that is almost a definition, and a weak point to make. Any signal that does not come from the pixels escapes a bound on the pixels. The real point is that provenance pays for this escape. A credential that is present and verifies is strong evidence. The absence of one tells us nothing, because most real content carries no credential and platforms strip the ones that exist (A5). So detection and provenance fail in opposite ways. Detection loses power as real and fake content grow alike. Provenance loses coverage as credentials go missing. Neither limit implies the other, and that is why both belong in the same stack.

**Limitations of the conjecture.** We note three caveats. First, the conjecture covers detectors that use the pixels alone. A detector that also uses metadata, provenance, or another out-of-band signal is not bound by it. Second, it is an asymptotic claim, not a statement about today. Current generators are still far from matching the real distribution, and detection remains useful against the generators in use now. Third, the conjecture is only weakly falsifiable, because TV cannot be estimated from samples in high dimensions. The clearest way to refute it would be a family of generators whose output stays easy to detect even as other measures of realism saturate. The practical takeaway is about where to invest. As generators improve, the returns to detection research should fall, while the returns to provenance should not.

## 9 Benchmarks and Evaluation

The empirical foundation of detection research rests on several benchmark datasets with distinct properties. FaceForensics++ (Rössler et al., 2019) provides 1,000 videos manipulated via four methods at three compression levels; XCEPTIONNET accuracy ranges from 99.26% (raw) to 81.00% (c40). Celeb-DF v2 (Li et al., 2020b) provides substantially higher visual quality; most detectors drop to 50–65% AUC, establishing it as the standard cross-dataset test. DFDC (Dolhansky et al., 2020) (Meta, $10M) provides 128,154 clips from 3,426 actors; the competition winner achieved only 65.18% on the black-box test. DF40 (NeurIPS 2024) covers 40 deepfake approaches including diffusion models, addressing the critical gap that pre-2023 datasets contain no diffusion-model deepfakes. Deepfake-Eval-2024 Chandra et al. (2025) provides the first large-scale in-the-wild benchmark: 44 hours of video, 56.5 hours of audio, and 1,975 images from 88 websites in 52 languages, documenting 45–50% accuracy degradation for deployed systems.

Yan et al. (2023) introduced DEEPFAKEBENCH at NeurIPS 2023, unifying 15 detection methods across 5 datasets in a standardized pipeline, addressing the critical problem of inconsistent evaluation protocols that made cross-method comparison unreliable. The AI-Face benchmark (Lin et al., 2025) provides the first million-scale demographically annotated dataset across 37 generation methods, enabling systematic fairness evaluation. VIGIL's OmniFake benchmark (2026) introduces a hierarchical 5-level evaluation from in-domain to in-the-wild social media data, progressively tested up to content from the latest generators (Nano Banana, Veo 3, Sora 2).

A critical gap persists: no benchmark simultaneously covers video, audio, text, and multimodal synthetic content at production quality, creating a blind spot for evaluating the comprehensive detection pipelines needed for the composed defense stack described in §7.

## 10 The Generalization Crisis

The composition analysis in §7 is motivated by the systematic failure of detection as a standalone defense. We summarize the six structural limitations.

**Cross-dataset collapse.** XCEPTIONNET drops from 99.26% on FF++ to 71.5% on Celeb-DF (28 pp). Deepfake-Eval-2024 Chandra et al. (2025) documents 45–50% degradation across 44 hours of in-the-wild content. Many off-the-shelf models produce AUC near 0.50.

**Adversarial attacks.** Carlini & Farid (2020) reduced a 0.95-AUC detector to 0.0005 (white-box) and 0.22 (black-box) with imperceptible perturbations altering as little as 1% of image area.

**Compression.** Social media platforms apply proprietary recompression that strips the high-frequency artifacts detection methods exploit. A deepfake detectable on one platform may evade detection after reprocessing by another.

**The diffusion barrier.** GAN-trained frequency detectors suffer approximately 15.2 pp AUROC drops on diffusion outputs Ricker et al. (2024). DF40 found detectors at 85% AUC on face-swap datasets dropping to 47% on Stable Diffusion outputs Yan et al. (2024).

**Demographic bias.** Lin et al. (2024) documented maximum false positive rate (FPR) gaps of 20.6 across intersectional subgroups. Ju et al. (2024) found Black men misclassified as deepfake at 39.1% versus white women at 15.6%.

**Human performance.** The meta-analysis of Diel et al. (2024), covering 56 papers and 86,155 participants, found average human accuracy of 55.54% (95% CI: 48.87–62.10%), not significantly above chance. Without being warned that a deepfake might be present, detection in realistic viewing conditions is poorer still, which leaves human judgment an unreliable safeguard and reinforces the case for the layered technical defenses this paper analyzes.

## 11 Societal Context

### 11.1 Scale of Harm

The scale of harm is now documented with specificity. Deepfake-enabled fraud drained an estimated $1.65 billion in 2025, part of $2.19 billion in cumulative losses by early 2026 Surfshark (2026), with the large majority occurring on social media. We note that these figures come from industry tracking rather than peer-reviewed measurement. Lalchand et al. (2024) projects AI-facilitated fraud losses reaching $40 billion by 2027 at a 32% compound annual growth rate. The Arup incident (World Economic Forum, 2025) (January 2024) saw a finance worker authorize $25.6M in transfers to deepfake participants in a multi-person video call. Non-consensual intimate imagery constitutes 96–98% of deepfake videos DeepStrike (2025), targeting women in over 99% of cases. Lakatos (2023) reported 24 million unique monthly visitors to 34 "nudify" service websites in September 2023. Insikt Group (2024) documented 82 political deepfakes across 38 countries from July 2023 to July 2024.

**The Deepfake-as-a-Service economy.** The threat has also industrialized. Ushakov (2025) collected over 300 Telegram and dark-web posts that advertise deepfake tooling for hire, sold between 2022 and 2025. Synthetic identity kits and per-video jobs go for tens of dollars. Once production is this cheap, only a systemic defense can match the scale, and that is the composed stack of §7. Agentic systems make this worse. They chain identity generation, voice cloning, and live video calls into one pipeline, and they are starting to run campaigns that used to need a person.

### 11.2 The Detection Paradox

A second-order harm is the *liar's dividend*: the ability of any actor to dismiss authentic evidence as AI-generated, exploiting the mere existence of synthesis capability. We have formalized this elsewhere, showing through a game-theoretic model that improving detector accuracy can paradoxically increase the payoff from false accusations of fabrication. The mechanism is the interaction between detector accuracy and challenge credibility: a more accurate detector makes the claim "this could be a deepfake" more persuasive. Provenance shifts this equilibrium by making the "claim it's fake" strategy dominated when cryptographic credentials exist.

### 11.3 Regulatory Context

The EU AI Act (European Union, 2024) (August 2024) requires AI output disclosure and machine-readable marking (Article 50, enforceable August 2026). The TAKE IT DOWN Act (119th U.S. Congress, 2025) (May 2025) criminalizes non-consensual intimate deepfakes in the U.S. China's Deep Synthesis Provisions (Cyberspace Administration of China, 2022) (January 2023) mandate visible labeling and real-identity

verification. 47 U.S. states have enacted deepfake legislation, with 174 total laws (82% passed in 2024–2025). The regulatory trajectory supports the composition thesis: multiple layers of technical defense are necessary because no single regulatory framework achieves global coverage.

**Dual-use acknowledgment.** We acknowledge that the Defense Composition Matrix and cascade failure analysis in §7 could be read as an instructional blueprint for bypassing current defense systems. We have deliberately focused on architectural patterns rather than implementation-specific attack code, and the vulnerabilities we document (metadata stripping, the analog hole, watermark removal) are already well-known in the adversarial ML and security communities. We believe the benefit of systematizing these failure modes for defense designers outweighs the marginal information gain for adversaries, who already exploit these weaknesses. The composition thesis itself is a defense-forward contribution: it argues for building layered systems that cover each other's blind spots, which is actionable only for defenders. We are equally wary of the opposite reading. The matrix is not evidence that current defenses are mature. The same analysis shows that even a full stack leaves residual exposure (A6, A7), and that most content today carries no protection at all. Overstating what these defenses can do invites a false confidence, and that is as damaging as the threat.

## 12    Open Problems

Our analysis identifies eight open problems, each stated as a falsifiable hypothesis.

**OP1: Architecture-agnostic detection.** *Hypothesis:* Foundation model representations encode generator-invariant facial properties that enable detection across all current architectures. *Protocol:* Evaluate GenD and CLIPping on outputs from five generators released after their training data was collected. If cross-generator AUROC exceeds 85%, the hypothesis is supported.

**OP2: Composition optimality.** *Hypothesis:* The optimal ordering of defense layers is provenance $\rightarrow$ watermark $\rightarrow$ detection, because provenance handles the broadest attack set and detection serves as backstop. *Protocol:* Deploy three orderings on a platform and measure false acceptance rates under the seven attack classes.

**OP3: Watermark-provenance integration.** *Hypothesis:* Integrating watermark status into the C2PA manifest (rather than treating them as independent systems) eliminates the "authenticated contradiction" vulnerability (E1). *Protocol:* Implement integrated and independent versions; measure user trust in the contradiction scenario.

**OP4: Calibrated detection for legal proceedings.** *Hypothesis:* Foundation model detectors with metric learning objectives produce better-calibrated confidence estimates than cross-entropy-trained models. *Protocol:* Evaluate expected calibration error across the foundation-model detectors surveyed in §5.

**OP5: Real-time detection at platform scale.** *Hypothesis:* Knowledge distillation from GenD-scale models to mobile-deployable architectures preserves >90% of cross-dataset AUROC at >100× throughput. *Protocol:* Distill and benchmark on the DFDC test set under latency and throughput constraints.

**OP6: Cross-modal joint detection.** *Hypothesis:* Audio-visual joint detection using foundation models outperforms modality-specific detectors because cross-modal inconsistencies provide a generator-invariant signal. *Protocol:* Evaluate on AVFakeBench and CharadesDF.

**OP7: Closing the analog hole.** *Hypothesis:* Imperceptible screen-embedded signals (e.g., modulated backlight patterns) can survive re-recording with a camera, providing partial defense against A6. *Protocol:* Embed and recover signals across five re-recording conditions.

**OP8: Demographic fairness in composed systems.** *Hypothesis:* Composing detection with provenance reduces demographic bias because provenance decisions are identity-independent. *Protocol:* Evaluate FPR disparity across demographic subgroups for detection-only versus detection+provenance systems.

## 13   Deployment Guide

The composition analysis points to a short rule for system designers, ordered by leverage. If you control the capture device, sign the content at capture with C2PA in hardware (Layer 1). This blocks A1 through A4 before any manipulation is possible. If you generate AI content, add two marks to every output. Embed a watermark in the content (SYNTHID or VIDEOSEAL) and attach a C2PA manifest. A later metadata strip (A5) then still leaves the embedded mark. The highest-value software step is to preserve the C2PA manifest through upload, transcoding, and delivery, since most of the residual exposure comes from platforms that strip it. For content that arrives with no provenance or watermark, run a foundation-model detector as a backstop, such as GEND at 0.03% parameter tuning. Set the threshold to the stakes. Use a strict false positive rate for forensic work and a looser one for triage. What remains is the analog hole (A6) and a compromised trust root (A7). No digital layer can close these. They call for media literacy and pre-bunking, not another mechanism.

## 14   Conclusion

The central finding of this analysis is that no single defense mechanism can address the AI-synthesized media challenge at the scale and sophistication it has reached. The Defense Composition Matrix reveals that of seven attack classes, only one (the analog hole) defeats every defense layer simultaneously, while infrastructure compromise bypasses the primary trust layer and metadata stripping is caught by the embedded watermark. The remaining attacks are blocked by at least one layer, meaning that a properly composed stack provides substantially greater resilience than any individual component.

Provenance emerges as the most resilient layer: it survives attacks A1 through A4, because cryptographic signatures are independent of pixel content. Detection, despite its dramatic improvements from foundation model approaches (cross-dataset AUROC improving from 71.5% in 2019 to 92.8% in 2025), faces a structural ceiling as generators approach the real data distribution. Watermarking occupies a complementary position, catching content that evades detection but vulnerable to removal attacks that detection can identify.

The practical implication is architectural. A researcher designing a platform's content authenticity system should use the Defense Composition Matrix to select layers that cover each other's blind spots. The eight open problems we identify, each with a falsifiable hypothesis and proposed experimental protocol, chart the path toward a defense stack whose residual vulnerabilities are minimized.

The window for building this infrastructure is finite. As AI-generated content grows from a minority to a potentially dominant fraction of online media, establishing provenance before that inversion is a categorically different engineering challenge than attempting to authenticate the majority of internet content retroactively.

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
