# OpenReview forum: "The Deepfake Defense Stack: Why No Single Layer Works and How They Must Compose"
_TMLR — Decision pending for TMLR_

### Review · Reviewer_5NWW · 2026-05-19

**Summary Of Contributions:**

This submission offers a practically important perspective on deepfake defense that detection, watermarking, and provenance must be evaluated as a composed stack rather than as isolated ones. The framing is useful and the paper is clearly written. However, the central composition matrix is not derived from a reproducible methodology and the argument frequently generalizes across modalities and threat models too aggressively, and the most formal claim in the paper is presented more strongly than the current evidence warrants.


**Strengths**

- The paper tackles an important question of how different defense layers interact under attack, rather than whether any one layer looks strong in isolation.
- The attack-centric organization is intuitive and practically useful. The composition matrix, cascade-failure framing, and deployment-oriented discussion make the manuscript easy to follow.
- The paper shifts attention from benchmark-centric detector comparisons to a broader systems view involving provenance, distribution shift, watermark removal, metadata stripping, and infrastructure compromise.

**Audience:**

Yes

**Audience Explanation:**

Researchers in deepfake detection, content authenticity, watermarking, and provenance would likely find the systems perspective valuable.

**Broader Impact Concerns:**

The paper is clearly defense-motivated. However, the attack taxonomy and failure analysis should be written carefully so they do not become overly operational guidance for adversaries. The paper should also avoid overstating the maturity of current defenses, since that could create false confidence in deployment settings.

**Claims And Evidence:**

No

**Claims Explanation:**

The main qualitative claim, that no single layer is sufficient, is convincing and well aligned with prior literature. The weaker point is methodological support for the stronger claims. The matrix compresses very heterogeneous evidence into crisp `B / D / X` labels without a reproducible synthesis procedure. The formal part is also not fully convincing. The Detection Ceiling Conjecture is interesting, but the theory presented is narrower than the prose claims, and the empirical support is still indirect. Overall, the paper’s intuitions are strong, but the evidence does not yet justify the same level of precision and generality used in the manuscript.

**Requested Changes:**

- The manuscript generalizes from mostly face/image/video evidence to the much broader category of AI-synthesized media. The paper needs either a narrower scope or modality-specific analyses instead of a single unified matrix.

- The core contribution, in Table 4, is not supported by a reproducible or auditable methodology. The paper states that the matrix covers 58 detection methods, 23 proactive defense systems, and 7 attack classes, but it does not provide the list of systems, an extraction protocol, an inclusion/exclusion rule, or a per-cell justification table.

- The treatment of watermarking is internally inconsistent. Section 5.3 discusses content-embedded watermarking systems such as SynthID and VideoSeal, but Table 4 marks `A5: Metadata strip` as a full bypass (`X`) for watermarking. That may be true for metadata-only marks, but it is not true for watermarking systems explicitly designed to survive metadata loss.

- The `Detection Ceiling Conjecture` is interesting, but it is currently closer to a position statement than to a well-supported technical result. The paper admits that current foundation-model detectors still achieve strong cross-dataset performance, which weakens the paper’s own move from observed detector brittleness to a structural inevitability claim. The manuscript should either substantially soften this section or support it with a clearer formalization and a more careful empirical argument.

- Table 3 is used to suggest a monotonic improvement from CNN methods to foundation-model methods, but the comparison is too neat for the evidence shown. These numbers come from different papers, different implementations, and potentially different evaluation pipelines.

---

> ### Author Response · Authors · 2026-05-30
> **Response to Reviewer 5NWW**
>
> We thank Reviewer 5NWW for this technical review. The strengths
> you noted are the ones we care about most, and the weaknesses you named were the
> right ones to fix. Each of them changed the manuscript.
>
> The watermarking inconsistency you caught was the sharpest single point, and it
> drove the largest structural change. You are correct that marking $A_5$, metadata
> strip, as a full bypass for watermarking is wrong for systems like SynthID and
> VideoSeal that are built to survive metadata loss. We split the watermark column
> into two, a metadata-bound mark and a content-embedded mark. Under the revised
> Table 4, $A_5$ bypasses the metadata-bound mark but is blocked by the embedded
> one. This corrected our headline finding as well. Once the column is split, the
> only attack that defeats every layer at once is the analog hole.
>
> The matrix methodology was the other core complaint, and we have made it
> auditable. We removed the "58 detection, 23 proactive, 7 attacks" counts, which
> implied a systematic denominator we had not earned. In their place, Section 2.1
> now states the scope and inclusion rule for what the matrix covers, and Table 5
> justifies every cell against a specific result in the cited literature. A reader
> can check each label against its source. The labels stay qualitative because the
> attack classes are not measured on one common scale across the three layers, and
> we say so rather than compressing heterogeneous numbers into a false precision.
>
> On scope, you are right that we generalized from face, image, and video evidence
> to synthetic media as a whole. We have pulled that back. The matrix is now scoped
> to the visual setting where the evidence actually lives, and where audio or text
> behave differently we flag it rather than folding it in. The comparative
> detectability table in Section 3.7 keeps the modalities distinct for the same
> reason.
>
> The Detection Ceiling needed the most care, and we softened it the way you
> suggested. We separated the two claims that were tangled together. The inequality
> is now stated only as an observation about balanced accuracy, with the explicit
> note that it does not carry over to AUC, since AUC ranks across all thresholds
> and can exceed the single-threshold bound. The structural-inevitability language
> is gone. What remains of the stronger claim is an empirical conjecture, marked as
> such, and we concede openly that total variation is not estimable in practice and
> that foundation-model detectors still generalize well across datasets. That last
> fact is exactly why we no longer call this an information-theoretic limit. It is
> an architectural one for now, and the abstract was changed to match.
>
> Table 3 was too clean, and we agree the numbers cannot bear a controlled reading.
> They come from different papers, implementations, and evaluation pipelines, and
> we now say so directly. The table is presented as indicative of a trend, not as a
> head-to-head comparison.
>
> On broader impact, we kept the attack taxonomy at the level of classes rather
> than operational recipes, and we strengthened the dual-use discussion. We also
> took care not to overstate how mature these defenses are, which is part of why
> the ceiling section is now stated more modestly.
>
> We are grateful for how precisely these points were made. They are the reason the
> paper is more honest about what it has shown.

---

### Review · Reviewer_DjzS · 2026-05-21

**Summary Of Contributions:**

The paper first summarizes recent advances in GenAI and investigates the current "crisis" of an increasing amount of undetectability of whether content was generated or not.
They categorize the resilience of three common techniques - passive detection, watermarking and content provenance -against 6 different adversaries. The main contribution is to interpret these three techniques as a stack that, combined, suggests giving stronger guarantees against any attacker. The paper indicates that there are some attack scenarios (A5 - A7) that can still bypass them simultaneously.

The paper does not provide its own empirical evaluations and relies on existing (published) literature.

**Audience:**

Yes

**Audience Explanation:**

Maybe, but I'm leaning towards no, as a) as a survey, this paper feels too incomplete, while b) the suggested classification lacks concrete justifications.

**Broader Impact Concerns:**

None.

**Claims And Evidence:**

No

**Claims Explanation:**

While I genuinely appreciate the author's effort in summarizing current advances in the field, I am not sure about the paper's main scope, and I feel like the paper is trying to push the frontier in too many places simultaneously.

It is explicitly not a survey or Sok although the categorizations in Chapters 3-5 indicate such a direction.
In my opinion, Table 4 summarizes the core contributions and observations of the paper. I like the message that even under combined defensive efforts, some attacks can still succeed. Unfortunately, I could not find any justifications for these claims or a concrete method for how they were evaluated.
Furthermore, I would need some justification for the term stack. Usually, this indicates some hierarchy, meaning that if one of them is bypassed, another layer can still act as a protection layer. Again, Table 4 shows that this is not the case. Essentially, the paper therefore summarizes which attacks are successful for which subset of defense mechanisms.

Finally, the state of the list of references is incomplete. Author lists are abbreviated; sometimes the journal is not listed. Some papers seem not to exist. By subsampling some random citations, I could, for example, not find the paper "Regeneration attacks remove invisible Watermarks" which was supposed to appear in NeurIPS.
"Invisible Image Watermarks Are Provably Removable Using Generative AI" (Xuandong Zhao, Kexun Zhang, Zihao Su, Saastha Vasan, Ilya Grishchenko, Christopher Kruegel, Giovanni Vigna, Yu-Xiang Wang, Lei Li).

**Requested Changes:**

- Reduce the paper to the necessary place. Section 3 can be shortened; maybe focus on the idea of these generative models.
- Focus on the core contribution, namely arguing that even combined defense mechanisms can still be bypassed.
- Fix the list of references, Fix the wrong citations.

---

> ### Author Response · Authors · 2026-05-30
> **Response to Reviewer DjzS**
>
> We thank Reviewer DjzS for a sharp read. Two of the points changed
> how we frame the paper, and one of them caught a real error we have since fixed.
>
> On scope, we agree we were reaching in too many directions, and we have narrowed
> it. The paper is no longer dressed as a survey. Sections 3 through 5 exist only
> to set up the composition analysis, so we compressed Section 3, cut the asides
> that were not load-bearing, and let Table 4 carry the argument. The core claim is
> the one you identified, that combined defenses still leave openings, and the
> revision puts it at the center.
>
> The term "stack" deserves a direct answer, because your reading is reasonable. We
> do not mean an ordered hierarchy where one layer fails over to the next. We mean
> defense in depth, layers with different coverage, so that an attack slipping past
> one is caught by another. The revised Table 4 is where this becomes visible, and
> it also corrects the version you read. After we split the watermark column into
> metadata-bound and content-embedded marks, $A_5$ no longer bypasses everything,
> because a screenshot strips the metadata but the embedded mark survives. Across
> the seven classes, only one, the analog hole $A_6$, defeats all three layers at
> once. Every other attack is blocked or at least degraded by some layer. So the
> matrix does not show that redundancy fails. It shows that redundancy holds for
> six of the seven attack classes, and that exactly one physical bypass escapes
> every layer. That asymmetry is the finding.
>
> On how the cells were obtained, this was a fair complaint and we have answered it
> concretely. We added Table 5, which justifies every cell against a specific
> result in the cited literature, so a reader can check each Blocks, Degrades, or
> Bypass label against its source rather than taking our word. The labels stay
> qualitative because the seven attack classes are not measured on one common scale
> across the three layers, and we now say so rather than implying a number we do
> not have.
>
> Finally, the references. You were right, and one of the entries you sampled was
> worse than incomplete. The citation to "Regeneration Attacks Remove Invisible
> Watermarks" did not correspond to a real paper. The work we meant is "Invisible
> Image Watermarks Are Provably Removable Using Generative AI" by Zhao et al.,
> NeurIPS 2024, and the entry now carries the correct title and full author list.
> After finding it, we did not stop there but we re-verified every cited reference
> against its official source. We treat this as the serious problem it is, and the
> bibliography has been rebuilt accordingly.
>
> We are grateful for the catch and for the framing advice. The paper is tighter
> and more honest for both.

---

### Review · Reviewer_qqqX · 2026-05-22

**Summary Of Contributions:**

This paper aims to do a survey on the attack/defense of deefake. According to the author’s description, it focuses on how several existing methods, namely, detection, watermarking and proactive defenses (based on crypto techniques), and whether or not it gives better results. There are also new conjectures and open questions proposed.

**Audience:**

Yes

**Audience Explanation:**

The topic of deepfake is well known, and discussing the progress of defense of deepfake is relevant to the area.

**Broader Impact Concerns:**

None.

**Claims And Evidence:**

No

**Claims Explanation:**

My main concern is the clarify. For example, the authors posed the key question in the intro:

“how do these defenses compose? When detection fails against a new generator, does watermarking catch the content? When a watermark is stripped by a screenshot, does provenance survive? When an adversary targets all three layers simultaneously, which attack strategies succeed and which are blocked by the redundancy of the stack?”

However, I don’t see the answers to these questions clearly presented, as a point to point manner, and hence not clear if these are answered. There are also other general aspects:

a) It does not seem to aim for a broader audience, and basic introduction to the problem scope is missing. When discussing concrete works, it talks about specific details rather than a organized high-level overview, which makes it less useful to non-experts.

b) Several claims without convincing evidence.

c) The writing style in many places are very informal, and does not look like an academic paper in computer science.

For detailed discussion of a) b) c), see my comments below.

- Sec 3.1, “FID’, “FFHQ” are mentioned but not defined/without citations

- The reference style is not consistent. For example, in the first paragraph of Sec 3.2 some paper is cited as “Rombach et al.’s Latent Diffusion (CVPR 2022),”, but the other places papers are cited as e.g., “ (Betker et al., 2023)”.

- Sec 3.2, what’s AUC? It’s mentioned here but I don’t see the definition

- Sec 3.3: a) there’s a long sentence about discontinuation of Sora — I think it covers way more details than needed, and whether or not the product is discontinued should not be significant in the discussion of methods/techniques in research. b) various time information was presented in this section (for example, Veo 3 May 2025), and it would be good to expand this to a citation? c) It’s also mentioned “The barrier to entry is now $20 per month” — I see this information hardly relevant to this paper.

- Sec 3.4, What does this mean: “MOS 4.53 vs. 4.58 for real speech”? In particular, what do the two numbers stand for, and what’s the meaning of MOS?

- Sec 3.4, there’s a paragraph about ElevenLabs, and there are a lot of discussions of the financial status of the company, which is largely irrelevant to this survey.

- You have a paragraph named “Detectability properties” in Sec 3.1 - 3.3, but it is absent in the remaining subsections of Sec 3. Why is this?

- Section 4.2, FF++ is mentioned in the first paragraph. This is the first time it is mentioned but I don’t see a definition/reference.

- Section 4, A1 A2 A3 are mentioned several times. But are they defined anywhere yet?

- Why isn’t “robustness under attack” paragraph provided for Sec 4.4 and Sec 4.6, whereas it is discussed in other subsections in Sec 4?

= In hindsight, I saw A1, A2… are defined only in Sec 6. One should define these before talking about it in e.g., Sec 4

- How are the information in Table 4 obtained? Are they from previous works? Also, are there concrete numbers than pass/degrade/bypass?

- In general, it seems the intro poses Sec 6 as a major contribution. However, I find Sec 6 overall not very comprehensive.

- There are around 60 - 80 references in total. To me, since the subject of the survey is multi-area of active study, this number of references is too small. I’m not an expert so I cannot say for sure you miss anything specific; however, having so few references might already defeat the purpose of writing a survey anyway: in my opinion, a survey needs to cover a significant portion in each touched area, which serves as a guide for people who are interested.

**Requested Changes:**

The detailed comments I listed above probably does not cover all issues, but they are representative. The authors should try to address the general issue beyond these specific comments. Overall, I think the paper should be rewritten.

---

> ### Author Response · Authors · 2026-05-30
> **Response to Reviewer qqqX**
>
> We thank Reviewer qqqX for a careful read. The clarity and evidence concerns
> were fair, and they shaped most of this revision.
>
> **Here's what has changed**
>
> The four questions we posed in the introduction now have a home. We added
> $\S7.3$, "Answering the Motivating Questions," and it takes each question in turn
> and points to the matrix cell that settles it. When detection fails against a new
> generator, an embedded watermark still survives, so that cell is a block. When a
> screenshot strips the metadata, the embedded mark survives but the bound
> provenance does not, so the two watermark columns diverge. When an adversary
> works all three layers at once, only the analog hole gets through every one of
> them. We should have answered these directly the first time, and now we do.
>
> We also agree the paper did not read for a broad audience, and we have changed
> its framing. It is no longer presented as a survey. It is a focused analysis of
> how the defense layers compose. A new Threat Model section ($\S4$) defines the
> attack classes $A_1$ through $A_7$ before any of them are used, which doubles as
> the high-level entry point a non-expert can read first. $\S3.7$ adds a
> comparative detectability table that gives the overview the method subsections
> were missing.
>
> Several terms were used before they were defined, and we fixed each one where it
> first appears. FID is now the Fr\'echet Inception Distance and FFHQ is
> Flickr-Faces-HQ in $\S3.1$. AUC is defined in $\S3.2$. MOS is the mean opinion
> score in $\S3.4$, and we now say plainly that the 4.53 is the score for synthetic
> speech against 4.58 for real speech. FaceForensics++ is spelled out and cited at
> first use. We also made the citation style consistent, so the Latent Diffusion
> reference now reads "(Rombach et al., 2022)" like every other citation rather
> than naming the venue in prose.
>
> The informal passages you pointed to are gone. We cut the Sora discontinuation
> discussion, the twenty-dollars-a-month aside, and the ElevenLabs financials. None
> of them carried the argument. Where a date still matters, it now has a citation.
>
> The uneven treatment across subsections is fixed in both directions. Every
> generation subsection in $\S3$ now carries a "Detectability properties"
> paragraph, and every detection subsection in $\S5$ now carries a "Robustness
> under attack" paragraph, including the two you noted were missing.
>
> On the composition matrix, we want to be clear about where the cells come from.
> They are not our private verdicts. We added Table 5, which justifies each cell
> against a specific result in the cited literature. The labels stay qualitative
> because the seven attack classes are not measured on one common scale across all
> three layers, and forcing a single number onto them would mislead. Where
> comparable numbers do exist, such as detection AUC under attack, we report them
> in the text.
>
> The reference count deserves a direct answer. We take the point, and it is the
> reason we stopped calling this a survey. A survey owes near-complete coverage of
> each area it touches. This paper owes a defensible account of how the layers
> interact, and its references are the set of methods that bear on that question.
> Table 1 places us next to the broad surveys that already exist, from Pei et al.,
> Deng et al., and Zhao et al., and shows what none of them do, which is to analyze
> how detection, watermarking, and provenance hold up together under attack and
> where the stack fails. That gap is the contribution.
>
> We removed the aggregate counts the abstract used to claim and verified every
> reference against its official source. We are grateful for the push on clarity,
> and we think the paper is stronger for it.